# Data-driven learning how oncogenic gene expression locally alters heterocellular networks

David J. Klinke II [1,2,3✉], Audry Fernandez[2,3], Wentao Deng[2,3], Atefeh Razazan[2,3], Habibolla Latifizadeh [4] & Anika C. Pirkey [1]

Developing drugs increasingly relies on mechanistic modeling and simulation. Models that capture causal relations among genetic drivers of oncogenesis, functional plasticity, and host immunity complement wet experiments. Unfortunately, formulating such mechanistic cell-level models currently relies on hand curation, which can bias how data is interpreted or the priority of drug targets. In modeling molecular-level networks, rules and algorithms are employed to limit a priori biases in formulating mechanistic models. Here we combine digital cytometry with Bayesian network inference to generate causal models of cell-level networks linking an increase in gene expression associated with oncogenesis with alterations in stromal and immune cell subsets from bulk transcriptomic datasets. We predict how increased Cell Communication Network factor 4, a secreted matricellular protein, alters the tumor micro-environment using data from patients diagnosed with breast cancer and melanoma. Predictions are then tested using two immunocompetent mouse models for melanoma, which provide consistent experimental results.

[1] Department of Chemical and Biomedical Engineering, West Virginia University, Morgantown, WV 26506, USA. [2] Department of Microbiology, Immunology and Cell Biology, West Virginia University, Morgantown, WV 26506, USA. [3] WVU Cancer Institute, West Virginia University, Morgantown, WV 26506, USA. [4] School of Mathematical and Data Sciences, West Virginia University, Morgantown, WV 26506, USA. ✉email: david.klinke@mail.wvu.edu

Tissues are dynamic structures where different cell types organize to maintain function in a changing environment. For instance, the mammary epithelium reorganizes during distinct stages of the ovarian cycle in preparation for lactation[1]. At the same time, immune cells clear dead cells and defend against pathogens present in the tissue microenvironment. Ultimately, the number and functional orientation of different cell types within a tissue interact to create a network, that is a heterocellular network. This heterocellular network is essential for creating and maintaining tissue homeostasis. While we know that tissue homeostasis is disrupted during oncogenesis, our understanding of how genetic alterations quantitatively and dynamically influence the heterocellular network within malignant tissues in humans is not well developed despite large efforts, like The Cancer Genome Atlas (TCGA), to characterize the genomic and transcriptomic landscape in human malignancy[2,3]. In parallel with these large scale data gathering efforts, two informatic developments, namely digital cytometry and Bayesian network inference, may be helpful in interrogating these datasets and are summarized in the next paragraphs.

In cytometry, single-cell sequencing technology elicits a lot of excitement as it enables unbiased discovery of novel cell subsets in particular disease states[4,5]. Unfortunately, persistent challenges related to confounding of batch effects with biological replicates limit the statistical power of these datasets to link oncogenic transcriptional changes with re-organization of the cellular network[6,7]. Due to the high number of biological replicates, transcriptomic datasets, such as the Cancer Genome Atlas, provide a rich resource in characterizing the heterogeneity of oncogenic transformation. Yet, these data were obtained from homogenized tissue samples and reflect the expression of genes averaged across a heterogeneous cell population. Computationally, "Digital Cytometry" can deconvolute the prevalence of individual cell types present within a mixed cell population[8]. The approach stems from the idea that the influx of a particular cell subset into a tissue corresponds to an increase in a gene signature uniquely associated with this particular cell subset[9–12]. Gene signatures of immune cells have been developed in a number of studies, which increasingly leverage scRNAseq data and machine-learning methods[13–16]. Besides representing different cellular subsets, gene signatures can also represent intracellular processes associated with oncogenesis, like the epithelial-mesenchymal transition[17–21]. Though, the predictive value of many of these tissue "features" in inferring how heterocellular networks are altered in diseased tissues remain unclear, as establishing correlations among features tends to be the end point of studies (e.g. refs. [19,22,23]).

Besides facilitating data acquisition, improved computational power has also enabled probabilistic inference methods that identify relationships within biological datasets that could not be observed using simpler statistical techniques[24,25]. These relationships can be depicted as a graph, where each node represents a random variable, or "feature", and an edge represents a direct relationship between two variables. When the direction of influence can be inferred for an edge, the parent-child relationship can be, under certain conditions, interpreted as causal and defined as an arc. A parent-child relationship implies that the values of these two random variables are not independent. By repetitively testing for independence conditioned on other subsets of variables, algorithms can learn the topology of a Bayesian network, which is expressed as a directed acyclic graph (DAG), directly from data[26]. As algorithms for reconstructing Bayesian networks emerged, they were used to model signaling pathways within cells[27], to identify known DNA repair networks in E. coli using microarray data[28] and to identify simple phosphorylation cascades in T cells using flow cytometry data[29,30]. While many more studies have been published since, a common conclusion is that the statistical confidence associated with an inferred network improves as the number of samples included in a dataset is greater than the number of random variables. However, transcriptomics data, like that obtained as part of the TCGA, typically have a large number of random variables ($n_{genes}$) and a small number of biological replicates ($n_{patients}$), which makes inferring gene-level networks computationally difficult and also implies greater uncertainty in the inference[31].

As summarized in Fig. 1, we propose an approach that combines digital cytometry with Bayesian network inference to identify how heterocellular networks associated with functional plasticity and anti-tumor immunity change during oncogenesis in humans. Conceptually, digital cytometry improves the statistical power by projecting the transcriptomic space onto a smaller number of "features" that estimate the prevalence of stromal and immune cell types and the average differentiation state of malignant cells present within the tumor microenvironment, such that $n_{features} << n_{patients}$. The causal structure among these features can then be predicted using Bayesian network inference. While data unstructured in time, such as the TCGA datasets, are not ideal for inferring causality, we test the inferred networks using in vivo experiments using syngeneic murine tumor models.

To illustrate the approach, we focused on Cell Communication Network factor 4 (CCN4/WISP1), a secreted matricellular protein that is upregulated in invasive breast cancer[32], expressed by malignant cells, and correlates with a lower overall survival in patients diagnosed with primary melanoma[33–35]. Functionally, expression of CCN4 promotes metastasis in melanoma by promoting a process similar to the epithelial-mesenchymal transition[33,34]. In developing state metrics that quantify functional plasticity in breast cancer and melanoma using an unsupervised approach, CCN4 was the only gene product associated with both a mesenchymal state metric in breast cancer and a de-differentiated state metric in melanoma that results in a secreted protein[21]. The collective set of features, or simply nodes of a network, were quantified in three transcriptomic datasets obtained from bulk tissue samples from patients with breast cancer and melanoma and used to generate a casual network describing how expression of a secreted gene product by malignant cells, such as from CCN4, more broadly alters the heterocellular network within a tissue using Bayesian network inference.

## Results

**Generating causal graphs linking oncogenes with heterocellular networks**. Bayesian network inference involves inferring the structure of the network, which captures the specific causal interactions or arcs among the nodes of a network and represents them as a directed acyclic graph (DAG), and then estimating the parameters of the conditional probability distribution from the datasets. As summarized in Supplementary Fig. 1, we used a four-step process to learn the causal structure associated with the cell-level networks. The four steps corresponded to specifying a "blacklist" based on prior information, generating an ensemble of potential arcs using 10 different structural learning algorithms, filtering potential arcs based on a trade-off between regression accuracy and model complexity to create a "whitelist", and learning the network structure using both the "blacklist" and "whitelist". We will discuss this process in more detail in the following paragraphs.

In learning the causal structure of a network, the network structure can be shaped by prohibiting the inclusion of specific arcs into a proposed network, that is by assigning an arc to a "blacklist" or alternatively a "no-list". Here, the "blacklist" represents a way to incorporate prior knowledge about causal

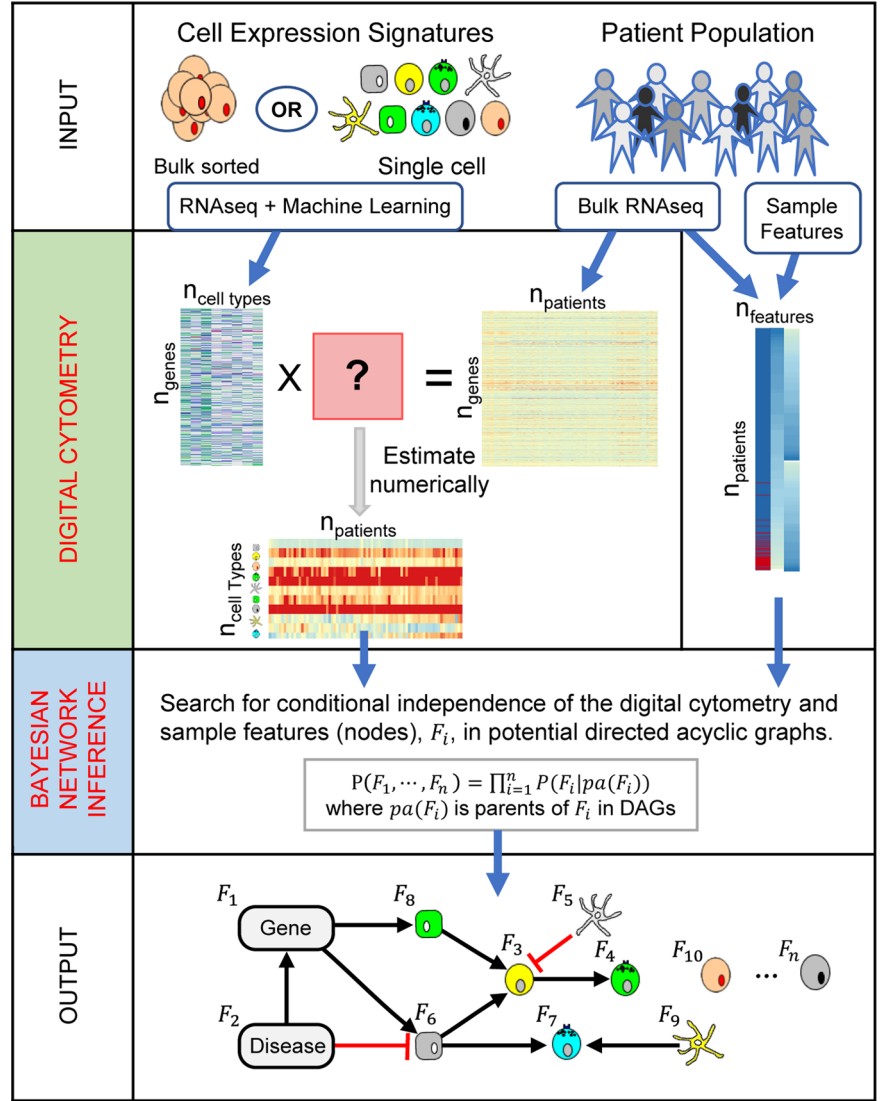

**Fig. 1 A computational workflow combining digital cytometry with Bayesian network inference to estimate how a genetic alteration associated with disease, such as over-expression of a secreted gene product by malignant cells, impacts the heterocellular network within a tissue.** Digital cytometry deconvolutes a bulk transcriptomic profile using gene signatures that correspond to different stromal, malignant, and immune cell types. The results estimate the prevalence of the different cell types within the tissue sample, that is the digital cytometry features. By using bulk transcriptomic profiles of defined patient populations, underlying variation in the inferred cellular composition coupled with features associated with a patient sample, such as over-expression of a secreted gene product by malignant cells, can be used to estimate how the heterocellular network is impacted by a genetic alteration intrinsic to the malignant cell using Bayesian Network inference. To illustrate the approach, we focused on malignant cell expression of Cell Communication Network factor 4 (CCN4), a secreted matricellular protein. The resulting directed acyclic graph represents the collective conditional independence among the modeled features, or nodes, of the network.

relationships associated with oncogenesis and the roles that specific immune cells play in controlling tumor cell growth. In particular, we considered only arcs into the "CD8 T cells" node (i.e., a leaf node), only arcs that originate from the "Cancer" node (i.e., a root node), mostly arcs that originate from the "CCN4" node (with exception for the "Cancer" node), and only arcs into the "CD4 T cells" and "Neutrophils" nodes. Cancer as a root node follows from contemporary understanding of oncogenesis, where mutation of either oncogenes or tumor-suppressor genes is the cause of cancer development[36]. Including expression of CCN4 as a root node follows from its frequent mutation in and scRNAseq-assayed expression by malignant cells[33,37]. Recent immunotherapies that either adoptively transfer CD8 T cells[38] or improve the function of CD8 T cells by relieving immune checkpoints[39] motivate specifying CD8 T cells as a leaf node. Specifying "CD4

T cells" and "Neutrophils" as leaf nodes follows from the high number of zero values for those features in the dataset, which were 350 and 439 samples in the BRCA dataset, respectively. These limits were implemented by assigning the corresponding arcs to the "blacklist". In a Bayesian context, assigning an arc to the "blacklist" specifies the prior probability of including this arc in network to zero. Practically, including these different categories of arcs into the "blacklist" helped infer arc direction consistently among the different structural learning algorithms (see Supplementary Fig. 2). As the number of arcs included in the "blacklist" was increased, the number of edges with unclear direction was decreased. Specifically, 17 edges had an unclear direction without specifying a "blacklist" (Supplementary Fig. 2A), while only 6 edges had an unclear direction in the final "blacklist" (Supplementary Fig. 2E).

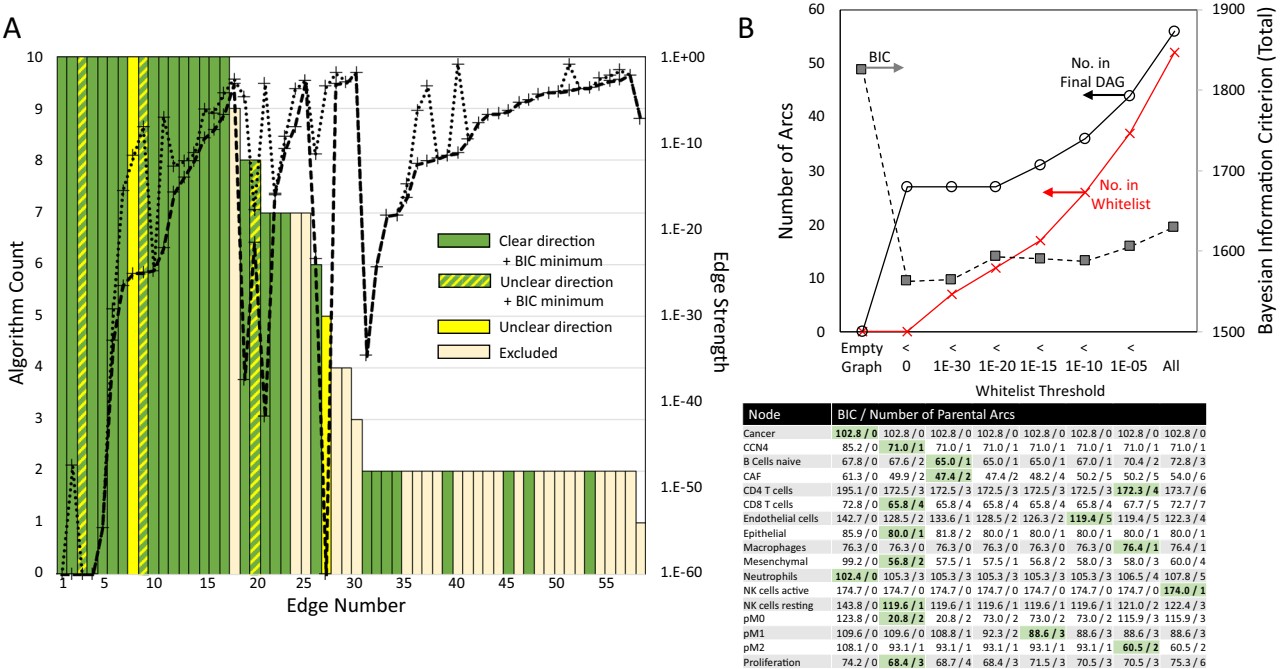

**Fig. 2 Summary of the evidence obtained from the TCGA breast cancer dataset supporting the consensus edges in the seed network. A** Edges ordered based on the number of algorithms that detected that an edge was enriched (bar graph - left axis) and the strength of enrichment (dotted lines - right axis). The strength of enrichment, that is Edge Strength, corresponds to the probability of a partial correlation between the two nodes of an arc being explained by random chance, given the rest of the network. The lines associated with the strength of enrichment represent the minimum (dashed line) and maximum (dotted line) values obtained by the different algorithms for each edge. Bar graph color indicates whether an edge was significantly enriched with a clear direction and contained within the set of arcs identified at the minimum BIC (green), significantly enriched without a clear direction but contained within the set of arcs identified at the minimum BIC (green/yellow), significantly enriched without a clear direction (yellow), or excluded from the consensus seed network list (tan). **B** Dependence of overall network connectivity (top) and node connectivity (bottom) on the number of arcs included in the whitelist. A threshold value for the edge strength (x-axis) was used to select arcs for including in the whitelist (red curve), which resulted in the inferred DAG connectivity (black circles). Values for the Bayesian Information Criterion (BIC) were calculated for the entire DAG (top - black squares) and for each node given the inferred parents (bottom table: BIC/number of parental arcs). The cells highlighted in green indicate the minimum BIC value and the number of corresponding arcs that were included in the consensus whitelist.

As algorithms for structural learning have different underlying assumptions, we used an ensemble approach to average across the different algorithms to identify a list of potential arcs that were used to seed the final learned structure of the DAG, that is a consensus seed network. This ensemble approach mirrors the "community network" approach described by Marbach and coworkers based on experiences with DREAM challenges for inferring intracellular networks[40,41]. Specifically, we used ten different structural learning algorithms[26], including constraint-based (Incremental association Markov Blanket - IAMB[42], Incremental association with false discovery rate control - IAMB.FDR[43], practical constraint - PC.STABLE[44], grow-shrink Markov Blanket - GS[45]), score-based (hill climbing - HC, Tabu search - Tabu[46]), and hybrid learning (max-min hill-climbing - MMHC[47], restricted maximization - RSMAX2[48]) algorithms. Two algorithms for local discovery of undirected graphs (max-min parents and children - MMPC[42], Hiton parents and children - SI.HITON.PC[49]) were also included to provide additional evidence supporting the existence of an edge between two nodes. Bootstrap resampling, that is resampling the dataset with replacement to generate a synthetic dataset of similar size as the original and infer the network structure using the synthetic dataset, was used in learning the network structure with each algorithm, which resulted in generating 10,000 network structures. For each algorithm, an averaged network structure was then calculated from this collection of network structures, where the threshold for including an arc into the average network was

automatically determined by each algorithm and was nominally 0.5 using the approach described by Scutari and Nagarajan[50]. We applied the same approach to both the breast cancer (BRCA - Figs. 2 and 3 and Supplementary Table 1) and the two melanoma datasets (common melanocytic nevi and primary melanoma: GEO - Fig. 5 and Supplementary Fig. 3 and Supplementary Table 2, and primary melanoma from the TCGA: SKCM - Fig. 5 and Supplementary Fig. 3 and Supplementary Table 3). Of note, edge/arc numbers are conserved across Supplementary Tables 1–3 and Fig. 2 and Supplementary Fig. 3.

The strength of evidence supporting the existence of an arc, that is arc strength, can also be used to filter arcs for inclusion in the consensus seed network, or simply called a "whitelist" or "yes-list". For instance, all arcs with a strength below a certain threshold can be included in the "whitelist" (see Fig. 2B). As this threshold was increased, more arcs were included in the "whitelist" and the number of arcs included in the final DAG was also increased. We also note that, while exploring the impact of a strength threshold, arcs were left out of the "whitelist" if their inferred direction varied among the algorithms (yellow bars in Fig. 2 and Supplementary Fig. 3). While increasing the number of arcs in the DAG better approximates the joint probability distribution, a complicated network limits interpretability. We used a Bayesian Information Criterion (BIC) to quantify this trade-off between regression accuracy and model complexity, where the optimal balance is at a minimum value. When the BIC was used to analyze the entire BRCA network, an empty graph

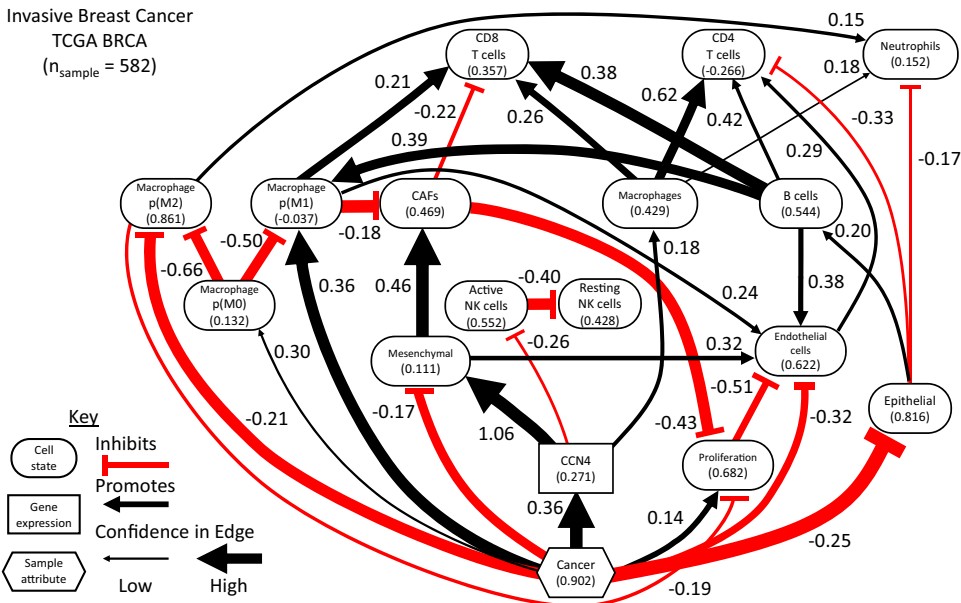

**Fig. 3 A directed acyclic graph (DAG) representing the conditional probability distribution inferred using the digital cytometry and sample features extracted from the breast cancer arm of the TCGA.** The nodes of the graph represent features, such as CCN4 gene expression (rectangle), sample attribute (hexagon), or the prevalence of a particular cell type/state (oval). The edges represent inferred causal relationships among the nodes. The black lines with arrow heads represent a positive causal relation while red lines with horizontal bars represent a negative or inhibitory causal relation, where the extent of influence of the parental node is annotated by the number beside the edge. The number included within the node symbol represents the average normalized value of the digital cytometry feature within the dataset with values of all of the parental nodes set to zero. The width of the edge is proportional to the posterior probability of inclusion into the DAG.

provided the maximum value and the minimum value corresponded to an empty "whitelist". As the weight of each node in contributing to the overall BIC value varied, we found that applying the BIC to individual nodes instead of the entire DAG provided better insight into including specific arcs in the final "whitelist". For instance, parental arcs associated with CCN4, Mesenchymal, and CD8 T cells were readily identified without specifying a "whitelist". Adding additional arcs with higher strength values (i.e., arc strength values greater than 1E-20) into the "whitelist" increased the BIC values. However, minimal BIC values for other nodes, such as active NK cells and p(M2), were only found when arcs with higher strength values were included. The arcs included in the "whitelist" used in the final analysis corresponded to the parental arcs associated with the minimum BIC values determined at the node-level for each dataset and that arcs didn't form cycles (see Fig. 2B and Supplementary Fig. 3). For instance, the minimum BIC value for the Endothelial node of 119.4 occurred in the BRCA analysis using a strength threshold for "whitelist" inclusion of 1E-10. This minimum BIC value was associated with five parental arcs: arc numbers 9 ("proliferation" → "Endothelial cells"), 19 ("Cancer" → "Endothelial cells"), 20 ("naïve B cells" → "Endothelial cells"), 33 ("Mesenchymal" → "Endothelial cells"), and 39 ("p(M1)" → "Endothelial cells"). However, only arc numbers 19, 33, and 39 were included in the "whitelist". Arcs 9 and 20 formed cycles with arcs identified at the minimum BIC values for other nodes and were excluded from the "whitelist". The final network for each dataset was generated using a hybrid learning algorithm (mmhc) using a "blacklist" specified based on prior causal knowledge and a "whitelist" corresponding to the consensus seed network. Similar to the first step, bootstrap resampling ($n_{boot}$ = 10,000) and network averaging were used to generate the three DAGs shown in Figs. 3 and 5. Each DAG was used to generate parameters for a linear Gaussian model estimated by maximum likelihood and conditioned on the network structure that approximates the joint

probability distribution associated with the dataset. Values for the linear coefficients and the average node values were used to annotate the DAGs. The sign of the linear coefficient was also used to annotate whether a particular arc promotes or inhibits the target node.

The resulting DAGs imply that oncogenesis in breast cancer was associated with a shift from epithelial to mesenchymal cell state accompanied by an increase in cell proliferation and a suppression of endothelial cells, which were inferred with high confidence. In turn, endothelial cells promote the infiltration of CD4 T cells. The local structure associated with "Cancer"'s influence on the "Mesenchymal" state via "CCN4" suggests an incoherent type-3 feed-forward motif to regulate the mesenchymal state. Inference of a feed-forward motif is interesting as feed-forward loops are highly prevalent and well understood as control mechanisms in intracellular networks[51] but are less well understood in the context of intercellular networks. In addition, expression of CCN4 inhibits active NK cells. The high confidence arc between active NK and resting NK cells follows from these features being mutually exclusive in the dataset and very few samples having zero values for both features. The mesenchymal state increased cancer-associated fibroblasts ("CAFs") with high confidence. Interestingly, oncogenesis was also associated with increasing the prevalence of a type 1 macrophage ("p(M1)"), which in turn promoted the recruitment of CD8 T cells. The prevalence of CD8 T cells are also connected to "Cancer" via a larger incoherent feed-forward motif involving "p(M1)", "CCN4", the "Mesenchymal" state, and "CAFs" with high confidence.

As there was more data supporting the BRCA DAG, the resulting Bayesian network model was compared against the underlying experimental data and used to explore the impact of varying CCN4 expression in the context of normal and cancer tissue (Fig. 4). To simulate normal and cancer tissue, we queried the conditional probability distribution by generating samples from the Bayesian network and filtered the values based on

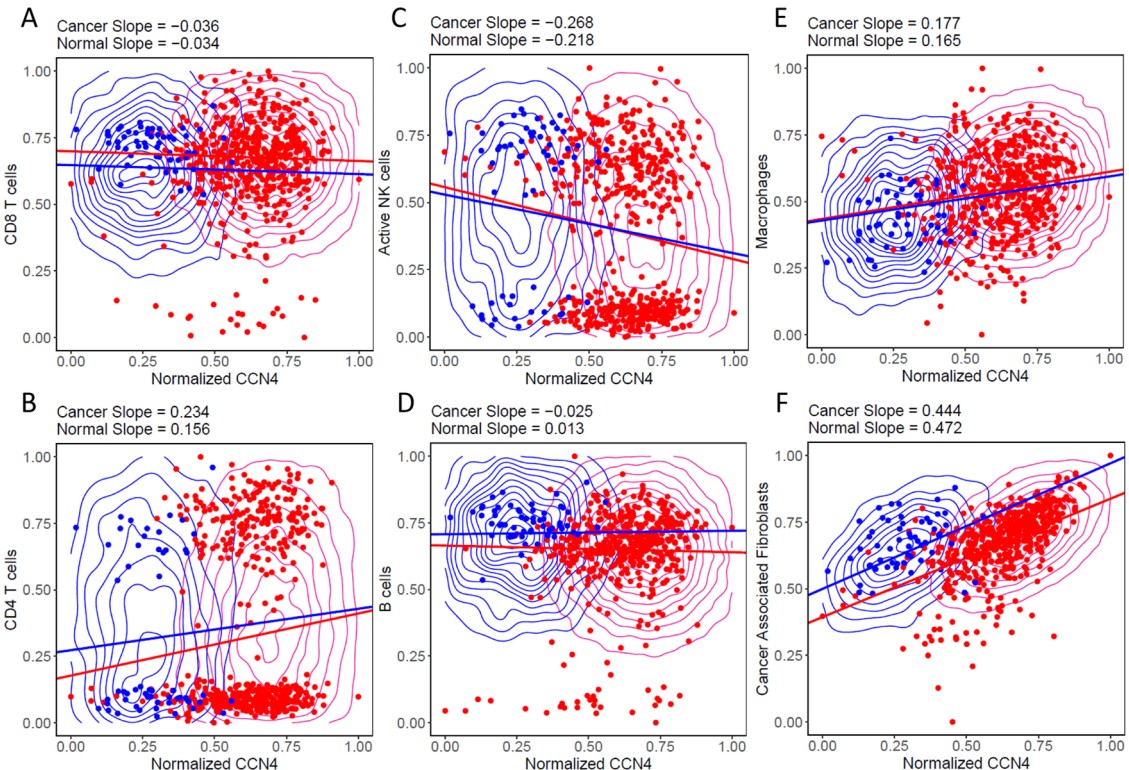

**Fig. 4 Conditional probability query of the BRCA DAG compared against digital cytometry estimates obtained from experimental data.** Experimental samples obtained from normal mammary and tumor tissue are shown as blue versus red dots, respectively. Samples of the conditional probability model for "Cancer" < 0.05 (blue contour) and "Cancer" > 0.95 (red contour) for CD8 T cells (**A**), CD4 T cells (**B**), active NK cells (**C**), B cells (**D**), Macrophages (**E**) and Cancer Associated Fibroblasts (**F**). Linear trend lines are superimposed on the conditional probability samples.

"Cancer" < 0.05 and "Cancer" > 0.95, which are colored in blue and red, respectively. The corresponding experimental data points and trend lines are overlaid upon the posterior distributions. The posterior distributions mirror the experimental data points, where there is an increase in CCN4 expression between normal ("Cancer" < 0.05) and cancer ("Cancer" > 0.95) tissue. The posterior distributions mirror the variability observed in the experimental data when comprised of non-zero values, such as CD8 T cells (Fig. 4A) and CAFs (Fig. 4F). In contrast, the prevalence of zero values increased the range of the posterior distribution, such as for CD4 T cells (Fig. 4B). In comparing normal to cancer tissue, CD8 T cells was the only feature, on average, increased in cancer tissue, while CD4 T cells and CAFs were decreased and active NK cells (Fig. 4C), B cells (Fig. 4D), and Macrophages (Fig. 4E) exhibited similar trends. Slopes of the trend lines highlight the influence of CCN4 expression on the prevalence of different immune cell populations. Increased CCN4 expression had the most pronounced inhibition on active NK cells and also suppressed CD8 T cells. CCN4 expression also had a pronounced positive impact on the prevalence of CAFs, macrophages, and CD4 T cells. CCN4 expression seemed to have no impact on B cells.

The breast cancer dataset contained 582 samples, of which 8.8% were from normal mammary tissue. In contrast, the two melanoma datasets contained 78 GEO samples, which includes 34.6% benign nevi, and 94 SKCM samples of primary melanoma only. While a lower number of samples limits the inferential power of a dataset, we decided to analyze them separately as they had different distributions in transcript abundance as a function of transcript length. As the Bayesian network inference algorithm leverages differences in the magnitude of a feature within a population, approaches to harmonize these two datasets may

introduce a systemic bias that is convoluted with oncogenic transformation, as the GEO dataset has many samples obtained from benign nevi while the SKCM dataset does not. In analyzing the melanoma datasets separately, we developed separate consensus seed networks using the node-level BIC values (see Supplementary Fig. 3). Using these consensus seed networks to specify the corresponding "whitelist", the structure and parameters associated with Bayesian network was inferred independently from each melanoma dataset (see Fig. 5).

Given the high prevalence of samples from benign nevi in the GEO dataset, high confidence arcs in the GEO network focus on changes associated with oncogenesis. Similar to the breast cancer analysis, oncogenesis was associated with a shift from an epithelial to a mesenchymal-like cell state. The mesenchymal cell state is promoted by both oncogenesis and CCN4 expression via a feed-forward motif, depicted here as the more common type 1 coherent motif. In addition, CCN4 expression indirectly impacted CAFs by promoting a mesenchymal-like cell state. Similar to the breast cancer analysis, oncogenesis promoted an increase in CD8 T cells, but indirectly by recruiting active NK cells. In analyzing the SKCM dataset, less emphasis is placed on the changes associated with oncogenesis but how expression of CCN4 influenced the network. Similarly to the GEO analysis, the SKCM analysis suggested that CCN4 expression directly impacted the mesenchymal state that then influenced CAFs. In addition, CCN4 expression directly promoted resting NK cells, which is similar to the BRCA analysis considering the reciprocal relationship between active and resting NK cells. In both melanoma datasets, CD8 T cells were directly promoted by macrophages and by NK cells either directly in the GEO dataset or indirectly through macrophages in the SKCM dataset. In addition, neutrophils, proliferation, and B cells were independent of all nodes. In all

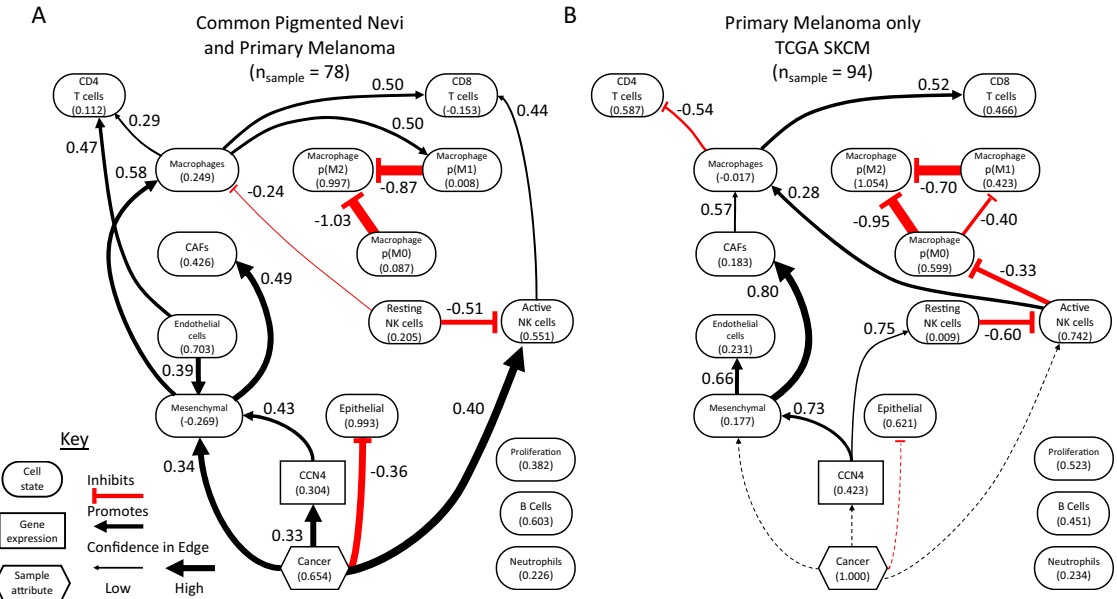

**Fig. 5 Two DAGs representing the conditional probability distributions inferred using the digital cytometry and sample features extracted from the two melanoma-related datasets. A** Analysis of a bulk RNAseq dataset obtained from patients with common pigmented nevi and primary melanoma ($n_{samples} = 78$). **B** Analysis of primary melanoma samples extracted from the SKCM arm of the TCGA ($n_{samples} = 94$). The DAGs are summarized using similar notation as described in Fig. 3. Dotted lines indicate edges that were included in the consensus seed network but, as the samples were all from patients with cancer, had no evidence in the TCGA dataset.

three analysis, there was high confidence associated with the arcs among the nodes quantifying macrophage polarization, which is likely an artifact of the formula used to calculate $p(M\Phi i)$'s (see Eqn 1). Queries of the conditional probability distribution based on the SKCM DAG for active NK cells, Macrophages, B cells, and CAFs were similar to the BRCA analysis (Supplementary Fig. 4). Similar to the BRCA analysis, a high number of zero values for the CD4 T cell features in the SKCM dataset suggests caution in interpreting differences in CD4 T cell predictions. To validate these predictions, we focused on two strategies: using flow cytometric analysis of tumor-infiltrating lymphocytes in syngeneic mouse tumor models to validate the digital cytometry predictions and targeted in vitro experiments to validate the presence or absence of arcs predicted by the Bayesian network inference.

**Validating the impact of CCN4 expression using syngeneic mouse models.** Syngeneic immunocompetent mouse models of cancer provide an important complement to retrospective studies of human data as they can aid in causally linking genetic alterations with cellular changes the tumor microenvironment. Here we used two syngeneic transplantable models for melanoma to test the predictions generated by the collective approach: the spontaneous B16F0 model and the YUMM1.7 model that displays Braf$^{V600E/WT}$ Pten$^{-/-}$ Cdkn2$^{-/-}$ genotype. As these cell lines basally produce CCN4 protein, we generated CCN4 knock-out (KO) variants using a CRISPR/Cas9 approach and confirmed CCN4 KO by testing conditioned media for CCN4 protein by ELISA. Tumors were generated by injecting the cell variants subcutaneously in 6–8-week-old female C57BL/6 mice and monitoring for tumor growth. Once wild-type (WT) tumors reached between 1000 and 1500 mm$^3$ in size, tumors were surgically removed from all mice that were not considered outliers and processed into single-cell suspensions ($n = 7$ for YUMM1.7 variants and $n = 4$ for B16F0 variants). The single-cell suspensions were aliquoted among three antibody panels to characterize the tumor-infiltrating lymphocytes by flow cytometry (see

Supplementary Figs. 5–7 for gating strategies). While the B16F0 and YUMM1.7 KO variants were generated using a double nickase CRISPR/Cas9 approach, similar results were obtained using a homology-directed repair strategy[34,35]. Additional controls for puromycin selection of CRIPSR/Cas9 edited cells using B16F0 cells transfected with a pBabe-puromycin retrovirus also behaved functionally similar in vitro and in vivo as wild-type B16F0 cells[34].

The percentage of CD45$^+$ cells among total live cells exhibited a semi-log dependence on tumor size (Fig. 6A - B16F0: $R^2 = 0.607$, F-test $p$-value $= 7.27E$-6; YUMM1.7: $R^2 = 0.830$, F-test $p$-value $= 1.48E$-7), where CCN4 KO resulted in smaller tumors in both cell models with greater CD45$^+$ cell infiltration. As illustrated in Fig. 6A, YUMM1.7 variants had a much higher dependence on tumor size than B16F0 variants. Conventionally, flow cytometry data are normalized to tumor size to estimate the prevalence of a particular cell type per tumor volume. Yet, the dependence on tumor size could be a confounding factor in addition to CCN4 expression that could skew the results. Moreover, the Bayesian network analysis predicts the impact of CCN4 expression alone on the prevalence of specific immune cell subsets. Thus, we focused instead on the prevalence of a particular cell type within the live CD45$^+$ TIL compartment to validate the digital cytometry predictions.

In comparing the WT B16F0 and YUMM1.7 models, the relative prevalence of NK (Live CD45$^+$ CD3e$^-$ NK1.1$^+$ B220$^-$ events), CD4$^+$ T (Live CD45$^+$ CD3e$^+$ CD4$^+$ CD8a$^-$ events), and CD8$^+$ T (Live CD45$^+$ CD3e$^+$ CD4$^-$ CD8a$^+$ events) cells were similar while B cells (Live CD45$^+$ CD3e$^-$ NK1.1$^-$ B220$^+$ events) were almost 10-times more prevalent in the B16F0 tumors compared to YUMM1.7 tumors (Fig. 6B). The prevalence of these different cell types changed within the CD45$^+$ TIL compartment upon CCN4 KO (Fig. 6C, D). Figure 6C highlights the trends among the mouse models and compares against the digital cytometry predictions obtained from the BRCA and SKCM datasets. Predictions for the change in cell type prevalence by CCN4 expression were obtained by comparing the prevalence of

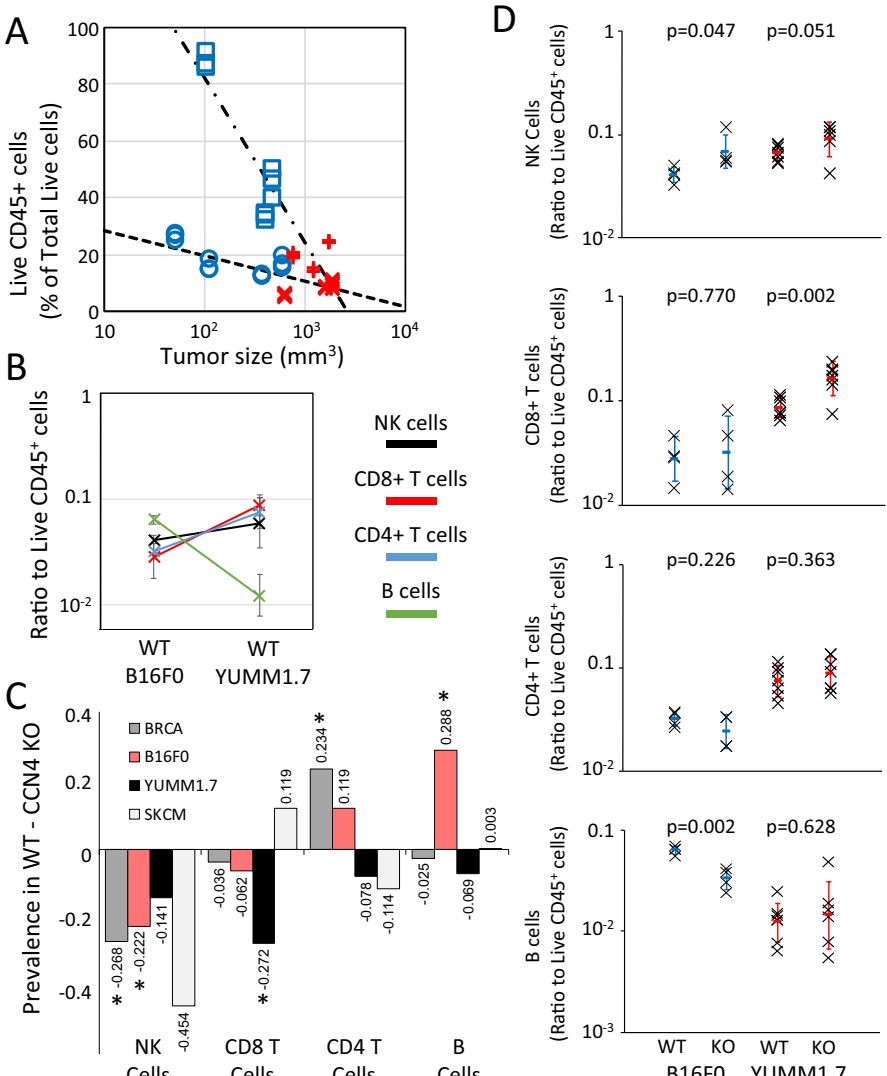

**Fig. 6 CCN4 knock-out in two syngeneic mouse models of melanoma induced a similar shift in cytolytic lymphocytes as observed in human breast cancer and melanoma. A** The percentage of live CD45+ cells isolated from tumors generated by inoculating s.c. with WT (red) and CCN4 KO (blue) variants of B16F0 (o and x's) and YUMM1.7 (□ and +'s) cells, where the log-linear trends are highlighted by dotted lines. CD45+ values were obtained from three different antibody panels that quantified T cells, B/NK cells, and myeloid cells in TIL isolates from each mouse. **B** A comparison of the ratio of NK cells (black), CD8+ T cells (red), CD4+ T cells (blue), and B cells (green) to live CD45+ TILs in s.c. tumors generated using WT B16F0 and YUMM1.7 cells (mean ± s.d.). **C** The difference in the mean prevalence of the infiltrating immune cell types was compared when CCN4 is present (WT) versus absent (CCN4 KO) as predicted by digital cytometry from the BRCA (dark gray) and SKCM (light gray) datasets and as observed experimentally using the B16F0 (red) and YUMM1.7 (black) mouse models. **D** TIL comparison upon CCN4 KO in B16F0 and YUMM1.7 mouse models stratified by NK cells, CD8+ T cells, CD4+ T cells, and B cells (top to bottom) ($n = 7$ biologically independent animals for YUMM1.7 and $n = 4$ biologically independent animals for B16F0 variants and mean ± s.d.). $p$-values calculated between WT and CCN4 KO pairs using two-sided Student's $t$ test.

the indicated cell type estimated by digital cytometry with CCN4 expression. Given the uncertainty associated with how particular tissue samples represent the full dynamic range of biologically relevant CCN4 expression, the BRCA and SKCM samples were separated into 5 quantiles based on CCN4 expression (see Supplementary Figs. 8 and 9). We used the feature values from the highest CCN4 expression quantile to represent wild-type tumors and from the lowest CCN4 expression quantile to represent CCN4 knock-out tumors. We then calculated the difference in cell type abundance upon CCN4 knock-out. Specifically, CD4 and CD8 T cells and B cells had analogous digital cytometry features as assayed in the flow panel, while NK cells were mapped to the "active NK cells" feature. The relative change in abundance was largely consistent among the four systems, with the YUMM1.7 model being the most different. The

BRCA and SKCM datasets predicted that NK cells were most reduced by CCN4 expression, which was observed in both the B16F0 ($p$-value = 0.047, $n = 4$) and YUMM1.7 ($p$-value = 0.051, $n = 7$) models. The BRCA dataset predicted that CCN4 expression reduced CD8+ T cells, which was observed in the YUMM1.7 model (YUMM1.7 $p$-value = 0.002) with a similar trend observed with the B16F0 model ($p$-value = 0.770). The CD4+ T cells seemed to vary in response to CCN4 expression as the BRCA and B16F0 results showed an increase while the SKCM and YUMM1.7 results showed a decrease. As stated previously, the BRCA and SKCM predictions for CD4 T cells should be interpreted with caution given the high frequency of zero values for the features. B cell response was mixed with both the BRCA and SKCM results suggesting no change. In mice, B cells were observed to decrease in the B16F0 model and increase in the

YUMM1.7 model upon CCN4 KO, with the low number of B cells infiltrating YUMM1.7 tumors rendered the results more variable. Given the small sample size of the experimental mouse cohorts, only the extremes were statistically significant, with NK cells significantly increased ($p$-value = 0.047) and B cells significantly decreased ($p$-value = 0.002) in B16F0 CCN4 KO tumors and CD8 T cells significantly increased ($p$-value = 0.002) in YUMM1.7 CCN4 KO tumors (Fig. 6D).

**CCN4 expression-induced changes in the myeloid compartment are less clear**. In addition to changes in T and NK cells within the live CD45$^+$ compartment, we also assayed myeloid subsets in tumors generated by WT and CCN4 KO variants of the B16F0 and YUMM1.7 cell lines. Using the gating strategy summarized in Supplementary Fig. 7, we focused on CD11c$^+$ and CD11c$^-$ macrophages (live CD45$^+$ CD11b$^+$ GR1$^-$ F4/80$^+$ events), neutrophils (live CD45$^+$ CD11b$^{int}$ CD11c$^-$ GR1$^+$ F4/80$^-$ events), dendritic cells (live CD45$^+$ CD11b$^{lo/int}$ CD11c$^+$ GR1$^-$ F4/80$^-$ events), and two different myeloid-derived suppressor cell (MDSC) subsets: CD11c$^-$ and CD11c$^+$ MDSC (live CD45$^+$ CD11b$^+$ GR1$^+$ F4/80$^{+/-}$ MHCII$^+$ events). In comparing tumors derived from WT cell lines, CD11c$^+$ macrophages were the most predominant infiltrating myeloid cell subset and most subsets were consistent between the two mouse models (Fig. 7A). Upon CCN4 KO in the mouse models, the CD11c$^+$ macrophage subset increased while the MDSC subsets decreased (Fig. 7B-E) within the CD45$^+$ compartment, while the response varied for the two least abundant subsets: neutrophils and CD11c$^-$ macrophages. The reduction in CD11c$^+$ MDSC in CCN4 KO variants were most pronounced and statistically significant ($p$ = 0.004 in YUMM1.7 and $p$ = 0.153 in B16F0). While Ly6G and Ly6C staining may have been a better staining strategy for distinguishing among monocytic (Mo-) and polymorphonuclear (PMN-) MDSC subsets, we observed a reduction in PMN-MDSCs in YUMM1.7 tumors upon CCN4 KO using Ly6G/Ly6C antibodies[35]. Consistent with the idea that PMN-MDSCs arise from impaired differentiation of granulocytes, neutrophils were increased within the CD45$^+$ compartment in CCN4 KO tumors derived from YUMM1.7 cells ($p$ = 0.002) but not statistically different in the B16F0 model ($p$ = 0.097). Other myeloid subsets trended similarly but with differences that were not statistically significant. In addition, we noted that a dendritic cell subset (live CD45$^+$ CD11b$^{lo/int}$ CD11c$^+$ GR1$^-$ F4/80$^-$ cells) increased upon CCN4 KO ($p$ = 0.045 in YUMM1.7 and $p$ = 0.011 in B16F0).

Comparing the trends in the myeloid compartment observed among the mouse models and the Bayesian network predictions obtained from the BRCA and SKCM datasets is less clear, given the uncertainty as to how the digital cytometry features map onto the quantified myeloid subsets in these mouse models. In particular, the digital cytometry features, that is the subset of particular genes and their relative expression that constitute a particular cell's gene expression signature, were defined using differentiated cell subsets. In contrast, MDSCs are prevalent within the tumor microenvironment and result from impaired differentiation of myeloid precursors into mature myeloid cells, like macrophages, dendritic cells, and neutrophils. Uncertainty in mapping how the gene expression signature in an immature myeloid cell subset overlaps with a differentiated myeloid cell creates uncertainty in predicting cell abundance by digital cytometry. Despite those concerns, key myeloid features in the Bayesian networks were macrophages oriented towards a M1 phenotype. Correspondingly, CD11c$^+$ macrophages, a subset that has been associated with pro-inflammatory M1 tumor-associated macrophages[52], were the most predominant myeloid subset in WT B16F0 and YUMM1.7 tumors and did not change upon

CCN4 KO. In the BRCA dataset, the prevalence of macrophages was influenced by CCN4 expression; yet, the functional orientation away from the M2 and towards the M1 phenotype depended solely on oncogenic transformation. Similarly, the prevalence of macrophages was influenced by both CCN4 expression and oncogenic transformation in both melanoma datasets. In contrast to the BRCA results, functional orientation of macrophages were independent of both oncogenic transformation and CCN4 expression. Neutrophils were predicted to be independent of CCN4 expression in the melanoma datasets, which is not surprising considering that the majority of tumors had zero values for the Neutrophil feature. Similarly, neutrophils were about 10 times less abundant than CD11c + macrophages in the mouse models. Given the significant changes observed in MDSCs and the corresponding differentiated cell subsets upon CCN4 KO in the mouse models, challenging digital cytometry predictions in this way highlights features that can be improved, such as discriminating among terminally differentiated and immature subsets, like Mo-MDSC and PMN-MDSC.

**Validating the predicted causal effects of CCN4 expression**. To validate the causal predictions present within the DAGs, we first focused on the inferred direct promotion of a mesenchymal state in malignant cells by CCN4 expression. To test this, we performed a rescue experiment with both CCN4 KO variants of the B16F0 and YUMM1.7 lines by adding recombinant CCN4 protein back to the cultures and monitored the expression of genes associated with the epithelial-mesenchymal transition in real time (Fig. 8A). Within 30 min of adding rmCCN4 to CCN4 KO YUMM1.7 cells, an increase in *Snai1* was observed and a concomitant reduction in the epithelial marker E-cadherin (*Cdh1*). We also observed similar dynamics using the B16F0 line. Collectively, these gene expression dynamics are consistent with CCN4 expression shifting malignant cells from an epithelial to mesenchymal-like phenotype.

While the focus typically is on the presence of an edge with the DAG, the absence of an edge can also provide information. In this context, we noted that local proliferation of CD8$^+$ T cells correlates with clinical response to immune checkpoint blockade[53,54]. In addition, the DAGs inferred from both the breast cancer and melanoma datasets suggest that a decrease in CD8$^+$ T cells is driven indirectly through CCN4 expression via modulating cancer-associated fibroblasts or the activity of NK cells. While the structural learning algorithms rejected a direct edge between CCN4 expression and CD8$^+$ cells, we tested whether CCN4 expression directly inhibits T cell proliferation (see Fig. 8B) using a statistical analysis of Cell Trace distributions in CD4$^+$ and CD8$^+$ T cells stimulated in vitro (see Supplementary Table 4). Specifically, splenocytes were stimulated in vitro with αCD3/αCD28-loaded beads in the presence of media conditioned by WT or CCN4 KO B16F0 cells or supplemented with 10 ng/ml recombinant mouse CCN4, which was consistent with the concentration in medium conditioned by WT cells. In both the CD4$^+$ and CD8$^+$ T cell populations, the presence of tumor-conditioned media significantly inhibited the fraction of cells that divided at least once (Dil - CD4 $p$-value = 0.022, CD8 $p$-value = 0.018) and the probability that a cell will divide at least once (PF - CD4 $p$-value = 0.024, CD8 $p$-value = 0.013) while CCN4 protein exposure was not a statistically significant factor. For responding cells, the average number of divisions they undergo (PI) was not different among experimental conditions for CD4$^+$ T cells ($p$-value = 0.22) but reduced in CD8$^+$ T cells exposed to tumor-conditioned media ($p$-value = 0.0077). Overall, the presence of tumor-conditioned media and not CCN4 protein influenced T cell proliferation, which was consistent with the DAGs.

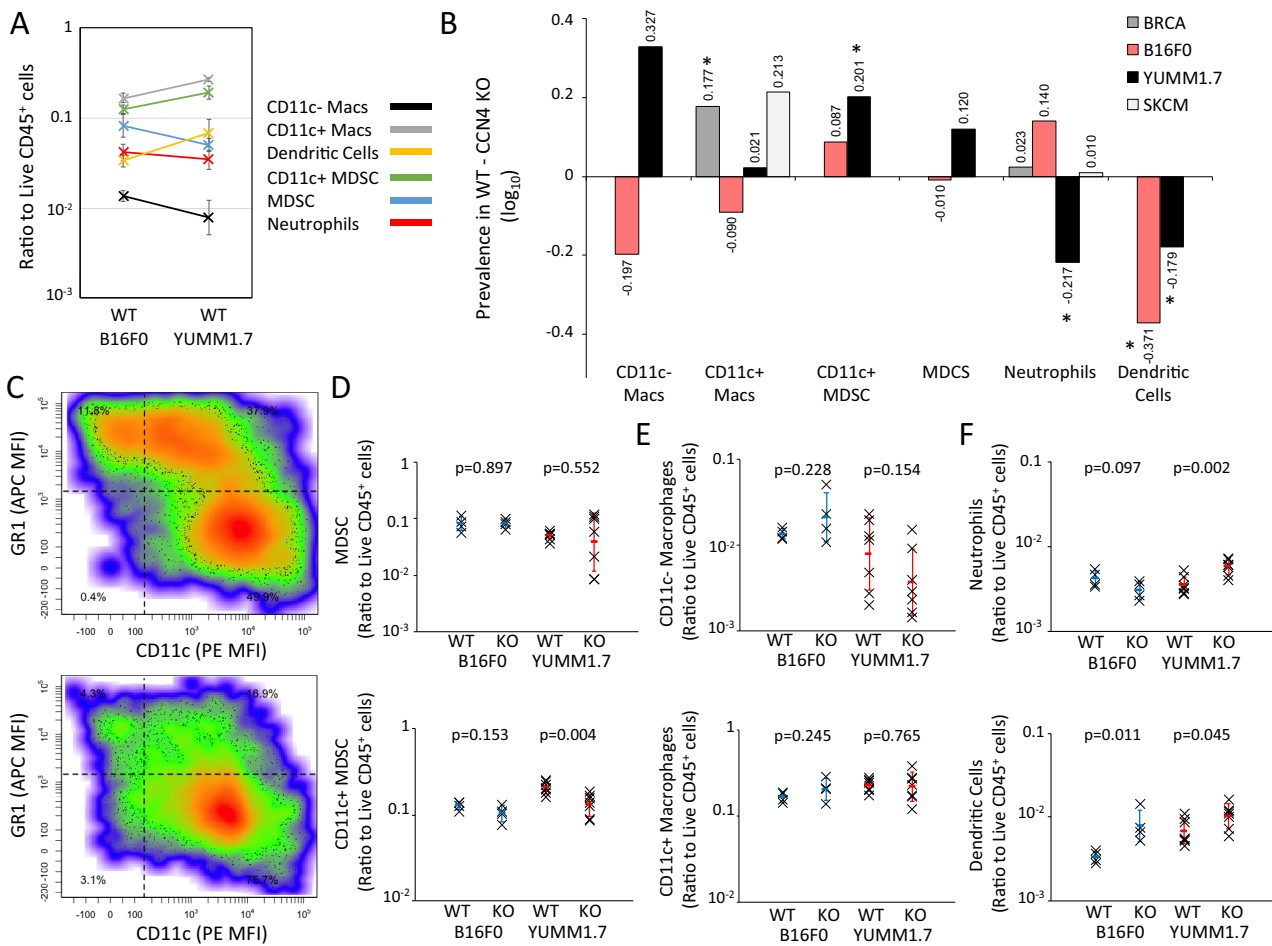

**Fig. 7 Myeloid immune cell subsets differentially infiltrated tumors derived from WT B16F0 and YUMM1.7 cells and shifted in prevalence upon CCN4 knock-out. A** A comparison of the ratio of CD11c- (black) and CD11c+ (gray) macrophages, Dendritic cells (yellow), CD11c+ MDSC (green), MDSC (blue), and Neutrophils (red) to live CD45+ TILs in s.c. tumors generated using WT B16F0 and YUMM1.7 cells (mean ± s.d.). **B** The difference in prevalence of the myeloid cell types was compared when CCN4 is present (WT) versus absent (CCN4 KO) as predicted by digital cytometry of the BRCA (dark gray) and SKCM (light gray) data sets and as observed experimentally using the B16F0 (red) and YUMM1.7 (black) mouse models. Macrophages are the only myeloid cell subset inferred from the BRCA and SKCM datasets and are assumed to be related to CD11c+ macrophages in mouse models. **C** A representative scatter plot of GR1 versus CD11c expression in gated live CD45+ CD11b+ TILs obtained from WT (top) and CCN4 KO (bottom) YUMM1.7 tumors. **D–F** TIL comparison upon CCN4 KO in B16F0 and YUMM1.7 mouse models stratified by myeloid-derived suppressor cell subsets (**D**: MDSC (top) and CD11c+ MDSC (bottom)) and other myeloid cell subsets (**E**: CD11c- (top) and CD11c+ (bottom) macrophages, **F**: neutrophils (top) and dendritic cells (bottom)) (n = 7 biologically independent animals for YUMM1.7 and n = 4 biologically independent animals for B16F0 variants and mean ± s.d.). p-values calculated between WT and CCN4 KO variants using two-sided Student's t test.

The third prediction tested was related to CCN4 expression either promoting resting NK cells in the primary melanoma DAG or inhibiting active NK cells in the breast cancer DAG. We note that the strong inhibitory edge between active and resting NK cells is a consequence of the mutually exclusive nature of these two nodes, as the weighting of many of the genes within resting and active NK cell gene signatures are similar (see Fig 8C). One of the most differentially weighted genes among these two gene signatures is IFNγ. NK cells also share cytotoxicity mechanisms and cytokine release, like IFNγ, with CD8+ T cells, Another characteristic of CD8+ T cells present within the tumor microenvironment is that they are dysfunctional[55]. As the digital cytometry approach used here doesn't estimate the functional state of CD8+ T cells only their prevalence within a tissue sample, we decided to test whether CCN4 expression had a direct impact on CD8+ T cell function, as quantified by target-specific ex vivo cytokine release as measured by ELISpot. First we generated YUMM1.7-reactive

CD8+ T cells by immunizing C57BL/6mice against YUMM1.7 cells and isolated CD8a+ T cells from splenocytes three days after re-priming with live YUMM1.7 cells. We also created a variant of CCN4 KO YUMM1.7 cells with CCN4 expression induced by doxycycline and vector controls that were used as target cells (see Supplementary Fig. 10). IFNγ ELISpots were used to quantify the CD8+ T cell functional response to the different tumor targets in the presence or absence of tumor-produced CCN4 protein (Fig. 8D). While doxycycline added in the context of a blank induction vector had no effect, re-expression of CCN4 protein by CCN4 KO YUMM1.7 cells following doxycycline induction significantly reduced IFNγ production (p-value < 0.002), which suggests that secreted CCN4 protein plays a direct role in inhibiting CD8a+ T cell function. Overall, the changes observed between WT and CCN4 KO variants of the B16F0 and YUMM1.7 mouse models were consistent with the causal networks inferred from the breast cancer and melanoma datasets.

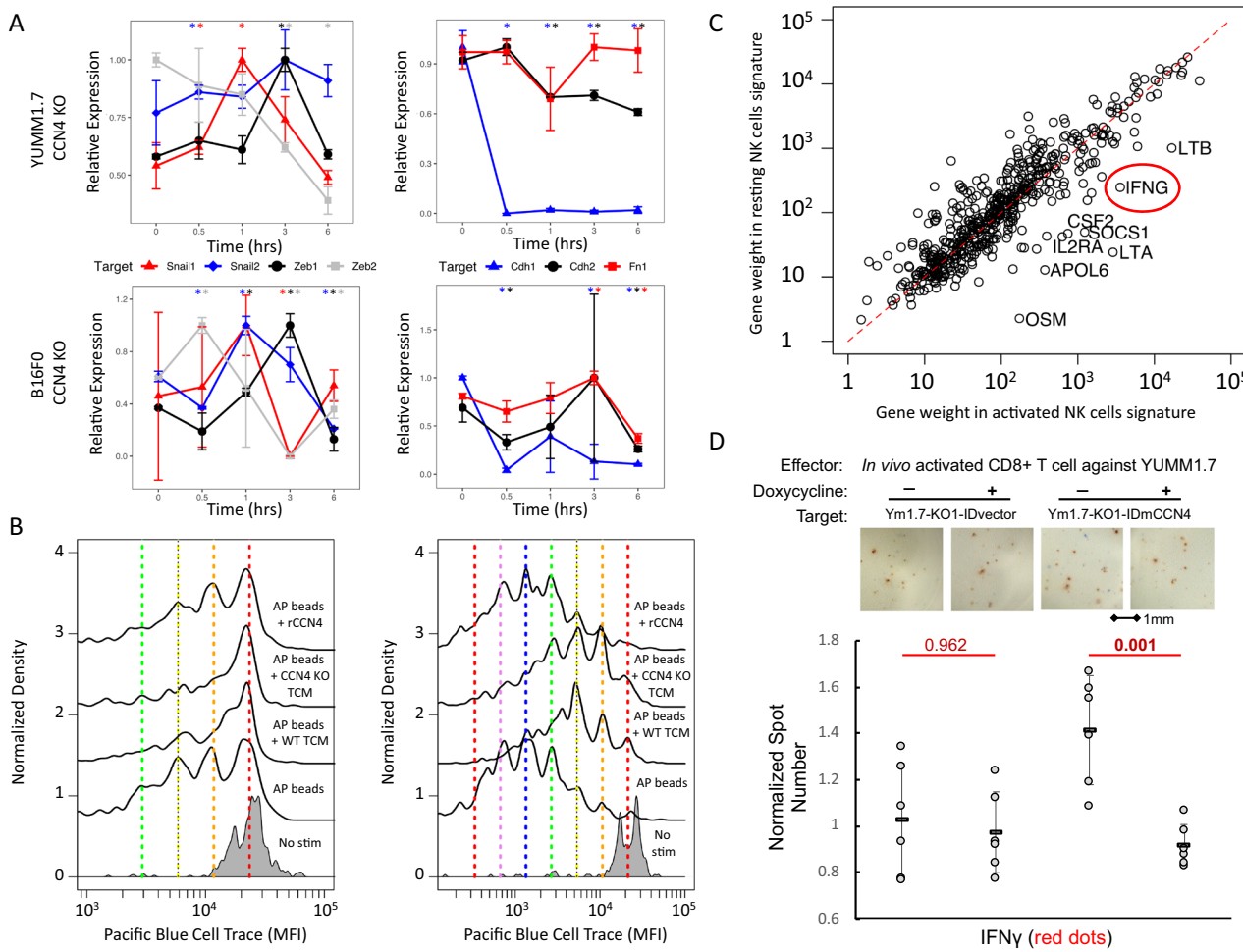

**Fig. 8 CCN4 promoted an epithelial-mesenchymal-like transition and had no direct effect on T cell proliferation but impaired CD8$^+$ T cell function.**
**A** Expression of genes for transcription factors (left panel - Snai1: red triangle, Snai2: blue diamond, Zeb1: black circle, and Zeb2: gray square) and adhesion proteins (right panel - Cdh1: blue triangle, Cdh2: black circle, Fn1: red square) associated with the epithelial-mesenchymal transition were assayed as a function of time following addition of rmCCN4 to CCN4 KO YUMM1.7 (top row) and CCN4 KO B16F0 cells (bottom row). Colored asterisks indicate whether gene at a particular time point was significantly different than untreated cells, where $n = 3$ biological independent samples. **B** The distribution in cell trace staining among live CD4$^+$ (left panel) and CD8$^+$ (right panel) T cells stimulated with $\alpha$CD3/$\alpha$CD28 (AP beads) alone or in the presence of media conditioned by WT B16F0 cells (AP beads + WT TCM), media conditioned by CCN4 KO B16F0 cells (AP beads + CCN4 KO TCM), or with 10 ng/ml of recombinant mouse CCN4 (AP beads + rCCN4). The distribution in the corresponding unstimulated cells (gray) are shown at the bottom. The colored vertical lines indicate the predicted dilution of cell trace staining in each generation based on the unstimulated controls. **C** Bivariate projection of the weights of genes within the resting (y-axis) and activated (x-axis) NK cell signatures. **D** Using spleens from C57BL/6 mice that were challenged with YUMM1.7 cells, isolated CD8$^+$ T cells were assayed by in vitro ELISpot for IFN$\gamma$ expression using variants of the YUMM1.7 cell line as targets (CCN4 KO YUMM1.7 with a blank inducible expression vector and CCN4 KO YUMM1.7 with a CCN4 inducible expression vector). To induce CCN4 expression, these YUMM1.7 variants were also cultured in the absence (−) or presence (+) of doxycycline and quantified following 24 h co-culture. Statistical significance between WT and CCN4 KO variants was assessed using two-way ANOVA followed by Tukey's multiple comparison ad hoc post-test, where $n = 6$ biologically independent samples. Results summarized as mean ± s.d.

## Discussion

Validating the role that a particular molecule plays in driving the disease state using targeted experiments is central for improving understanding of biological mechanisms or selecting among competing drug targets. Given the limited observability of the biological response in experimental models and patients, mechanistic modeling and simulation is playing an increasing role in helping answer many central questions in discovering, developing, and receiving federal approval of pharmaceutical drugs and also basic biology[56]. In immuno-oncology, there is increasing interest in modeling the heterocellular network of relevance for a specific immunotherapy. The first step in creating mathematical models of cell-level networks is to create the topology of the network, which is expressed in terms of which

nodes to include and how they influence each other. The structure of these cell-level models is created using a fully supervised approach, which means by hand using expert knowledge[57]. For instance, systems of ordinary differential equations have been developed to capture multiple spatial compartments containing interacting malignant, antigen presenting, and T cells and to predict a general immune response[58], a response to immune checkpoint blockade using CTLA-4, PD-1, and PD-L1 antibodies[59] or adoptive cell transfer[60].

While leveraging the knowledge of experts is a great starting point, hand-curated models can also implicitly impose bias on how data is interpreted. In the context of molecular-level networks, rules and algorithms have been developed to elaborate causal networks based on a limited set of rules[61–64]. The rules

constrain the types of interactions, or arcs, that are realistic between the nodes while the algorithms generate all possible arcs that are consistent with the rules and collection of nodes. The resulting rule-based networks are then used to interpret data by filter the arcs for the most consistent and, in the process, may reveal previously unappreciated pathways. For instance, a rule-based model was used to interpret single-molecule detection of multisite phosphorylation on intact EGFR to reveal a new role for the abundance of adapter proteins to redirect signaling[65]. Given the challenges with representing the various activation states of a 12-subunit $Ca^{2+}$/calmodulin-dependent protein kinase II (CaMKII) holoenzyme that is essential for memory function, a rule-based model identified a molecular mechanism stabilizing protein activity that was obscured in prior reduced models[66]. Inspired by engineering better CAR T cells, Rohrs et al. developed a rule-based model to interpret site-specific phosphorylation dynamics associated with Chimeric Antigen Receptors[67].

To our knowledge, no equivalent approach exists in the context of modeling cell-level networks. One might consider agent-based or cellular automata models to apply as the cellular interactions are specified by rules. In rule-based modeling of molecular networks, the rules and algorithms elaborate a network space that encompasses all possible topologies of the network and data is used to prune the network to the most relevant. Similarly, the arcs included in the "blacklist" and "whitelist" can be considered as a Bayesian prior, where the strength of inclusion in the final DAG and the coefficient associated with a particular arc in the conditional probability function depend on the data. In contrast, agent-based or cellular automata models require specifying all interactions between cells as rules a priori and are validated qualitatively by comparing emergent behavior against experimental observations[68–70]. We posit that coupling digital cytometry with Bayesian network inference is analogous to rule-based modeling in the context of modeling cell-level networks. Here, the rules comprise a limited set of constraints, or heuristics, related to the direction of information flow. Specifically, the rules limit how changes in gene expression within the malignant cell introduced during oncogenesis propagate to stromal and immune cells present within the tumor microenvironment and are implemented as a "blacklist". In formulating the "blacklist" used here, specifying cancer as a root node and $CD8^+$ T cells as a leaf should be generalizable in the context of most solid tumors. Due to immune privilege, glioblastoma multiform may be an exception for $CD8^+$ T cells[71]. Specifying the oncogene as a child of cancer and a potential root node to everything else should also be generalizable. Choosing an oncogene that produces a secreted product is also important as the secreted product is likely to play a direct role in intercellular communication. Single-cell RNAseq data can help support this assumption that malignant cells express the oncogene. Specifying nodes that have a high percentage of zero values as leaves follows from network inference arguments as trends in the non-zero values, which ideally are related to other nodes, may be swamped by differences between the zero and non-zero values and may give rise to spurious arcs. The algorithms that underpin Bayesian network inference search over all possible network topologies for arcs that are consistent with the data. The resulting networks can be used in multiple ways. As an unsupervised approach, the network topology could complement existing workflows for creating mechanistic mathematical models fit for use in testing molecular targets[57,72]. In addition, DAGs represent explicit hypotheses generated from pre-existing human data that motivate experiments to validate the predictions, as illustrated by the B16F0 and YUMM1.7 results.

While the focus here is in the context of breast cancer and melanoma due the pre-existing breadth of data, the approach could be generally applied to other biological contexts and motivate experimental studies. For instance, one of the limitations of inferring the network topology in the form of directed acyclic graphs is that some direct and indirect causal relationships can be confounded, such as reciprocal feedback modes of communication between cells[73]. Discerning the difference between a direct and indirect causal relationship has practical importance, such as for selecting therapeutic targets[74]. Methods, like Granger causality and dynamic Bayesian networks[31,75,76], do exist that could reveal direct and indirect causal relationships, but time-series data is required. Unfortunately, human tissue samples, like those in the TGCA, are very rarely sampled with time. Analysis of pre-existing human datasets can be complemented by a more focused experimental study of a pre-clinical model. Specifically, single-cell RNAseq to identify the cell types present and their associated gene signatures can be combined with bulk transcriptomic sequencing to capture the prevalence of all of the cell types within the tissue sample and provide a large number of biological replicates spanning the disease space - normal homeostasis; initiation; early, middle and late progression; and productive resolution or adverse outcomes. Similar network topologies would suggest similar biological mechanisms and help select relevant pre-clinical models for drug development. In short, we feel that combining digital cytometry with Bayesian network inference has the potential to become an indispensable unsupervised approach for discovering relevant heterocellular networks associated with disease.

## Methods

The research complies with all relevant ethical regulations. In particular, the West Virginia University Institutional Review Board classified all analysis of existing data obtained from humans as exempt from Human Subjects Research (IRB Protocol #1604090889). In addition, all animal experiments were approved by West Virginia University (WVU) Institutional Animal Care and Use Committee (IACUC Protocol #1604002138) and performed on-site.

**Digital cytometry**. Transcriptomics profiling of bulk tissue samples using Illumina RNA sequencing for the breast cancer (BRCA) and cutaneous melanoma (SKCM) arms of the Cancer Genome Atlas were downloaded from TCGA data commons, where values for gene expression were expressed in counts using the "TCGAbiolinks" (V2.8.2) package in R (V3.6.1) and converted to TPM. RNA-seq data assayed in samples acquired from benign melanocytic nevi and untreated primary melanoma tissue and associated sample annotation were downloaded from GEO entry GSE98394 and converted to TPM. TCGA data and the benign nevi and melanoma data were filtered to remove sample outliers and normalized based on housekeeping gene expression[77]. Digital cytometry features associated with the functional plasticity of malignant cells within an epithelial to mesenchymal-like state space were calculated based on state metrics developed separately for bulk breast cancer and melanoma tissue samples[21]. Cell proliferation features were calculated based on the median expression of genes associated with cell proliferation that were identified previously using human cell line data[34]. Features corresponding to the prevalence of endothelial cells, cancer-associated fibroblasts, macrophages, and $CD4^+$ T cells were calculated using CIBERSORTx (https://cibersortx.stanford.edu) using the gene signatures derived from single-cell RNAseq data[37] while the prevalence of B cells naïve, $CD8^+$ T cells, Macrophage M0 ($M\Phi0$), Macrophage M1 ($M\Phi1$), Macrophage M2 ($M\Phi2$), activated NK cells, resting NK cells, and neutrophils were calculated using the LM22 immune cell gene signatures in CIBERSORTx run in absolute mode.

Given the potential lack of independence among the macrophage features, the LM22 macrophage features were combined to estimate the probability of the average functional orientation using the formula described previously[71]:

$$p(M\Phi i) = \frac{M\Phi i}{M\Phi 0 + M\Phi 1 + M\Phi 2}, \tag{1}$$

where $i = \{0, 1, 2\}$ and denotes the specific macrophage subtype. Additional cellular features were excluded from the analysis as they tended to have a large number of zero values across the datasets, which resulted in being disconnected from the rest of the network in preliminary structural inference. Sample attributes were transformed to numerical values, which were assumed to be extremes of a continuous variable. For instance, if the sample was obtained from normal tissue, the value for "Cancer" was set equal to 0; if the sample was obtained from cancer tissue, the value for "Cancer" was set equal to 1. The sample attributes, CCN4 gene expression, and estimated cellular features extracted from the bulk RNAseq data calculated for each sample are included in the GitHub repository.

**Bayesian network inference**. Prior to network inference, feature values were log transformed, normalized to values between 0 and 1, and discretized (BRCA: 15

intervals; GEO and SKCM: 6 intervals). The features were then assigned to nodes. The relationships among the nodes, or arcs, were represented by directed acyclic graphs inferred from the datasets using a process involving four steps, as detailed in the results section and graphically summarized in Supplementary Fig. 1. In learning the network structure, the strength of evidence supporting the existence of an arc, that is arc strength, corresponds to the probability of a partial correlation between two nodes of an arc being explained by random chance, given the rest of the network. Specifically, arc strength is the p-value calculated using the exact t-test for a Pearson's correlation coefficient. Given the inferred structure, a Bayesian network in the form of a linear Gaussian model was fit to the datasets using maximum likelihood estimation of the model parameters. A Bayesian Information Criterion (BIC) was used to evaluate the trade-off between a L1 loss function, which numerically quantifies the ability of a specified directed acyclic graph to represent the conditional independence of the joint probability distribution, and a penalty associated with network complexity. A BIC value for feature $j$ was calculated by:

$$BIC_j = n_{arcs} \cdot log_{10}(n_{obs}) + \sum_{i=1}^{n_{obs}} |\hat{Y}_{ij} - Y_{ij}|, \quad (2)$$

where $\hat{Y}_{ij}$ and $Y_{ij}$ are the observed and predicted values for feature $j$, $n_{arcs}$ is the number of parental arcs associated with the entire DAG or node, and $n_{obs}$ is the number of samples. Conditional probability queries of the Bayesian networks were performed by logic sampling with $10^5$ samples. Bayesian network inference was performed using the 'bnlearn' package (V4.5) in R (V3.6.1).

**Reagents and cell culture.** Cytokines and antibodies were obtained from commercial sources and used according to the suppliers' recommendations unless otherwise indicated. The mouse melanoma line B16F0 (purchased in 2008, RRID: CVCL_0604) and HEK293T (purchased in 2005, RRID: CVCL_0063) were obtained from American Tissue Culture Collection (ATCC, Manassas, VA). The mouse melanoma line YUMM1.7 (received in September 2017, RRID: CVCL_JK16) was a gift from Drs. William E. Damsky and Marcus W. Bosenberg (Yale University)[78]. HEK293T, B16F0 and YUMM1.7 cells were cultured at 37°C in 5% $CO_2$ in high-glucose DMEM (Cellgro/Corning) supplemented with L-glutamine (Lonza), penicillin-streptomycin (Gibco), and 10% heat-inactivated fetal bovine serum (Hyclone). All cell lines were revived from frozen stock, used within 10-15 passages that did not exceed a period of 6 months, and routinely tested for mycoplasma contamination by PCR. CCN4 knock-out variants of B16F0 and YUMM1.7 cells were generated using a double-nickase CRISPR/Cas9 editing strategy described previously[33]. Briefly, two pairs of mouse CCN4 double nickase plasmids that target the mouse *Ccn4* gene at different locations were purchased from Santa Cruz Biotechnology, Inc. (Dallas, TX) and transfected into B16F0 and YUMM1.7 cells following the manufacturer's instructions. Following antibiotic selection, surviving single clones were isolated and expanded on six-well plates. The concentration of CCN4 protein in the cell culture media from those wells was assayed using the Human WISP-1/CCN4 DuoSet ELISA Kit (R&D Systems, Minneapolis, MN) to confirm CCN4 knockout. CCN4-knockout cells were further expanded and aliquoted to create a low passage frozen stock.

**In vivo tumor assays and in vitro T cell proliferation assays.** C57BL/6Ncrl mice (6–8-week-old female) were from Charles River Laboratories. Mice were randomly assigned to treatment groups and co-housed following tumor initiation. Animals were housed with a 12-h light/dark cycle (light 6 a.m. to 6 p.m.), temperature nominally 74 degrees F, and humidity 50%. Subcutaneous tumors were initiated by injecting mice subcutaneously with $3 \times 10^5$ of the indicated YUMM1.7 cells and $2.2 \times 10^5$ of the indicated B16F0 cells in 100 μL and, once palpable, tumor sizes were recorded every other day via caliper. Tumor volume was calculated using the formula: $0.5236 \times width^2 \times length$, where the width is the smaller dimension of the tumor. Once WT tumors reached between 1000 and 1500 $mm^3$ in size, the tumors were surgically removed from mice in both arms of the study (WT and CCN4 KO) after euthanasia and processed into single-cell suspensions. This normally occurred at Day 14 with the B16F0 model and at Day 27 with the YUMM1.7 model. Seven tumors were processed separately for each YUMM1.7 variant while four tumors were processed for each B16F0 variant. Single-cell suspensions were obtained by enzymatically digesting the excised tumors using the Tumor Dissociation Kit and gentleMACS C system (Miltenyi Biotec, Auburn, CA). In addition to following the manufacturer's instructions, the gentleMACS program 37C_m_TDK_1 was used for B16F0 tumors and 37C_m_TDK_2 was used for YUMM1.7 tumors. Following lysing of the red blood cells, the remaining single-cell suspensions were washed and stained with Live/Dead Fixable Pacific Blue Dead Cell Stain Kit (ThermoFisher). Following blocking with Mouse BD Fc Block (purified rat anti-mouse CD16/CD32 antibodies, BD Biosciences), the surface of the cells were stained with one of three different antibody mixes that focused on T cells (CD45, CD3, CD4, CD8, and PD1), NK and B cells (CD45, CD3, B220, NK11, DX5, and PD1), and myeloid cells (CD45, CD11b, CD11c, Gr-1, F4/80, and MHCII) and quantified by flow cytometry. The specific antibodies and dilutions used are listed in Supplementary Table 5.

To assess the impact of CCN4 protein on T cell proliferation in vitro, splenocytes were obtained from naïve C57BL/6 mice and stained with CellTrace Pacific Blue Cell Proliferation Kit (ThermoFisher). Stained splenocytes ($2.5 \times 10^5$) were stimulated for 3 days in 96-well plate with MACSiBeads loaded with anti-

mouse CD3 and anti-mouse CD28 antibodies (AP beads, Miltenyi Biotec), at a 1:1 proportion. Fresh serum-free DMEM media conditioned for 24 hours by either confluent wild-type (WT TCM) or confluent CCN4 KO (CCN4 KO TCM) melanoma B16F0 cells were collected, centrifuged to remove cells and cell debris, and added at 50% final volume during T cell stimulation with AP beads. In addition, splenocytes were either left unstimulated or stimulated with AP beads alone, or stimulated in the presence of recombinant mouse CCN4 protein (rCCN4, R&D) at a final concentration of 10 ng/mL, which was selected based on the concentration of CCN4 observed in medium conditioned by WT cells. Each experimental arm was performed in biological triplicate. After 72h, cells were washed and stained with Live/Dead Fixable Green Dead Cell Stain Kit (ThermoFisher). Surface staining with anti-mouse CD8/APC (Miltenyi Biotec), anti-mouse CD4/APC-Cy7 (BD Biosciences), anti-mouse CD62L/PE (eBioscience, ThermoFisher) and anti-mouse CD44/PerCP-Cy5.5 (eBioscience, ThermoFisher) was performed after incubating the cells with Mouse BD Fc Block (BD Biosciences). The proliferation of both CD4+ and CD8+ T cells were quantified by flow cytometry. Results representative of one of two independent experiments.

**In vitro suppression of CD8+ T cell function.** Inducible mouse CCN4 expression lentiviral vector (IDmCCN4) was constructed with Gateway cloning using Tet-on destination lentiviral vector pCW57.1 (Addgene Plasmid #41393, a gift from David Root) and pShuttle Gateway PLUS ORF Clone for mouse *Ccn4* (GC-Mm21303, GeneCopoeia). Lentiviruses were produced by transfecting either pCW57.1-mCCN4 or pCW57.1 and two packaging plasmids, psPAX2 (Addgene plasmid 122260) and pCMB-VSG-G (Addgene plasmid 8454), into HEK293T cells. Virus soup was aliquoted and used to transduce YUMM1.7 *Ccn4* CRISPR knockout (Ym1.7-KO1) cells with the expression vector[33]. After puromycin selection, two pools of cells with inducible mCCN4 (Ym1.7-KO1-IDmCCN4) or vector control (Ym1.7-KO1-IDvector) were obtained. ELISA tests with doxycycline (Dox, final 0.5 μg/ml) induction revealed the mCCN4 expression was under stringent control and the secreted protein was in the similar level as compared with wild-type YUMM1.7 cells (see Supplementary Fig. 10).

To generate YUMM1.7-reactive CD8+ T cells, healthy C57BL/6Ncrl mice were inoculated subcutaneously with irradiated YUMM1.7 cells ($10^5$/mouse), followed by live YUMM1.7 cells ($3 \times 10^5$/mouse) 3 weeks later. The mice without tumor growth in the next five weeks were maintained. Three days before the assay, the mice were injected again with live YUMM1.7 cells ($10^5$/mouse). On the day of assay, these mice were euthanized and the YUMM1.7-reactive cells were isolated from mouse splenocytes using mouse CD8a+ T Cell Isolation Kit (130-104-075, Miltenyi Biotec), resuspended in a concentration of $10^6$/ml. In total, 50 μl ($5 \times 10^4$) of the YUMM1.7-reactive CD8+ T cells were aliquoted into each well on a 96-well plate for ELISpot assay using Mouse IFNγ/TNFα Double-Color ELISpot kit (Cellular Technology Limited, CTL) following manufacturer's instructions. Briefly, target tumor cells were stimulated with IFNγ (200 U/ml or 20 ng/ml) for 24 h, harvested and resuspended in a concentration of $2 \times 10^6$/ml. In all, 50 μl ($10^5$) of indicated tumor cells in triplicates were aliquoted into each well, with or without doxycycline (Dox, 0.5 μg/ml). The reactions were incubated at 37°C for 24 hours and colored spots were developed (red for IFNγ and blue for TNFα). The spots were counted and imaged using an Olympus MVX10 Microscope and the result was plotted and analyzed by Microsoft Excel (version 16.57).

**RNA analysis.** All samples for RNA analysis were prepared in biological triplicates using CCN4 KO cells plated on 6-well plates in complete growth medium for 48 h before starting the time-course experiment with a wash and replacement of medium. In groups treated with recombinant mouse CCN4 protein (rmCCN4, 1680-WS-050,R&D systems), rmCCN4 was added at a final concentration of 5.0 μg/mL. At the indicated time points, cells were lysed and total RNA was isolated using the GeneJET RNA purification kit (Thermo Fisher Scientific). 50–500 ng of each RNA was reverse-transcribed using the High Capacity RNA-to-cDNA Kit (Thermo Fisher Scientific). Real-time quantitative RT-PCR was performed on a StepOnePlus real-time PCR system with Brilliant II SYBR Green qPCR master mix (Agilent Technologies, Santa Clara, CA). Glyceraldehyde-3-phosphate dehydrogenase served as the internal control for the reactions, and the results were analyzed in R to obtain normalized gene expression values. The primer pairs for the indicated genes in this work were adopted from PrimerBank[79] and listed in Supplemental Table 6.

**Flow cytometry.** Single-cell suspensions described above were stained with specific antibodies or isotype controls using conventional protocols. Fluorescence-activated cell counting was performed using a BD LSRFortessa and FACSDiva software (V8.0 BD Biosciences), where the fluorescence intensity for each parameter was reported as a pulse area with 18-bit resolution. Unstained samples were used as negative flow cytometry controls. Single-stain controls were used to establish fluorescence compensation parameters. For TIL analysis, greater than $5 \times 10^5$ events were acquired in each antibody panel in each biological replicate. In analyzing enriched cell populations, $2 \times 10^4$ events were acquired in each biological replicate. Flow cytometric data were exported as FCS3.0 files and analyzed using R/Bioconductor (V3.6.1), as described previously[80]. The typical gating strategies for T cells, NK and B cells, and myeloid cells are shown in Supplementary Figs. 5–7,

respectively. The statistical difference in tumor-infiltrating lymphocytes between WT and CCN4 KO variants was assessed using log-transformed values and a two-tailed homoscedastic Student's $t$ test. Cell proliferation was quantified using four metrics: fraction diluted (Dil), Precursor frequency, %dividing cells (PF), Proliferation index (PI), and proliferation variance ($SD^{D})[81]$. Statistical differences among these proliferation parameters were assessed using type III repeated measures ANOVA in the "car" (V3.0-7) package in R. A $p$-value < 0.05 was considered statistically significant.

**Statistics and reproducibility**. As this was an exploratory study, no sample-size calculation was performed prior to the study. However, independent experiments using multiple replicates are used to ensure reproducibility. The size of a cohort within a biological replicate was limited to the bandwidth of sample processing workflow. Mice were excluded from the study if they failed to develop tumors following subcutaneous challenge. Experiments were repeated at least once, or data were compiled from two independent experiments. Replicates were reproducible. Mice were purchased from the indicated vendors, labeled, and randomized to treatment groups/cages. Mice receiving different cell lines were housed in the same cages, at a density of five mice per cage. The investigators were not blinded to the group allocation during data collection or analysis, as the same individuals that set up the experiments were the ones that analyzed the results. They did remain objective in interpreting the data.

**Reporting summary**. Further information on research design is available in the Nature Research Reporting Summary linked to this article.

## Data availability

The key datasets used in the analysis can be obtained from the following sources:
• The datasets supporting the conclusions of this article are available in Gene Expression Omnibus repository with the following GEO accession number: GSE98394.
• Transcriptomics profiling of bulk tissue samples using Illumina RNA sequencing for the breast cancer (BRCA) and cutaneous melanoma (SKCM) arms of the Cancer Genome Atlas were downloaded from TCGA data commons, where values for gene expression were expressed in counts, using the "TCGAbiolinks" (V2.8.2) package in R (V3.6.1).

## Code availability

The code used in the analysis can be obtained from the following GitHub repository:
•https://github.com/KlinkeLab/CellNetwork_2020[82]

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

## Acknowledgements

This work was supported by National Science Foundation (NSF CBET-1644932 to D.J.K.) and National Cancer Institute (NCI 1R01CA193473 to D.J.K.). The content is solely the responsibility of the authors and does not necessarily represent the official views of the NSF or NCI. We also used equipment from the WVU Flow Cytometry & Single Cell core, which was supported by the National Institutes of Health Grants GM103488/RR032138, GM104942, GM103434, and OD016165.

## Author contributions

Conceptualization: D.J.K; Study design: D.J.K; Data acquisition: D.J.K., A.F., W.D., and A.R.; Data analysis: D.J.K., H.L., and W.D.; Data interpretation: D.J.K.; Funding acquisition: D.J.K.; Methodology: D.J.K.; Project administration: D.J.K.; Software: D.J.K. and A.C.P.; Supervision: D.J.K.; Writing - original draft: D.J.K.; Writing - review and editing: all authors.

## Competing interests

The authors declare no competing interests.
