## [Peer Review File · Nature Communications]

Reviewers' Comments:

Reviewer #2:

Remarks to the Author:

The authors answered the concerns raised by this reviewer and two other reviewers in the response letter. The authors conducted additional experiments to validate the computational predictions. This reviewer could find the biological importance of CCN4 to the regulation of immune cell population changed associated with EMT in mouse model (by validation) and that digital flow cytometry has contributed for successful identification of immune cell populations from TCGA bulk cell datasets. I also understand that the authors made substantial efforts to improve the manuscript for this revision.

However, the whole manuscript is still not convincing to me.

The biggest concern of this reviewer is the generality of this analysis method. A method of predicting cellular communication networks that combines digital flow cytometry and Bayesian network inference seems to have potential for wide use if this method is generally applicable to other types of bulk sequence datasets. However, as I wrote in my last review, I'm still not sure how to choose CCN4. This impression is mainly due to the lack of information on (1) the CCN4 level (protein or mRNA) of the original TCGA dataset, and (2) how to create "blacklists" and "whitelists". There is no such information in Method section. If CCN4 is not highly expressed in the datasets (BRCA and SKCM), or if there is no relationship between mRNA levels and malignancies (or prognosis) of the cancer, it is difficult to interpret this study focuses on CCN4-centric networks. Such data analysis is necessary at the beginning of the study. Creating blacklists and whitelists requires clear criteria that other researchers can follow and reproduce the method. In addition, data variation in each time point in Fig8 is too large to conclude the clear result. For these reasons, I am afraid to say that this manuscript lacks strong evidence of conclusions.

Response to Reviewer's Comments

Data-driven learning how oncogenic gene expression locally alters heterocellular networks.

David J. Klinke II, Audry Fernandez, Wentao Deng, Atefeh Razazan, Habibolla Latifizadeh, and Anika C. Pirkey

Submitted for review in Nature Communications (Manuscript ID: NCOMMS-21-07831C)

Overall, we thank the reviewers for their service and their constructive criticisms on the manuscript. We feel that this revised paper presents, in a more comprehensive way, our findings related to inferring how genetic alterations associated with oncogenesis alters heterocellular networks within tissues.

Our responses to the specific comments are listed below in blue font. A red font is used to indicate the text that has been changed in this revised manuscript.

Editorial Comments:

You will see that the reviewer continues to raise substantive concerns, shared by Reviewer #1 from the last round of review, that cast doubt on the advance your findings represent over earlier work and the strength of the novel conclusions that can be drawn at this stage.

Authors Response (AR): Regarding the advance over prior work, such as the high profile studies of Bendall et al. Science 2011 and Sachs et al. Science 2005, we **first** note that these focus on **intracellular** networks, that is the interactions among proteins within a studied cell type. Here, we are focused on **intercellular** networks, that is the interactions among different cell types within a particular tissue. **Second**, we note that using multiple algorithms and including the Bayesian Information Criteria are additional technical advances over these prior studies. The **third** point of difference related to earlier work is in relation to the application area. Here, we are focusing on the pharmaceutical interest in immuno-oncology, which inherently focuses on intercellular networks. We note that, in 2019, immuno-oncology became the leading category of pharmaceutical agents in RnD and is one of the most competitive. Mechanistic math modeling and simulation is becoming increasingly important in pharma RnD (see NIH white paper: <https://www.nigms.nih.gov/training/documents/systemspharmawpsorger2011.pdf>). Towards this end, Certara created a consortium with abbvie, astellas, Bayer, Merck, Shire and Takeda to create **by hand curation** an immuno-oncology simulation platform. https://www.certara.com/pressreleases/certara-launches-industry-first-quantitative-systems-pharmacology-qsp-consortium-on-immuno-oncology-with-leading-pharma-company-members/?ap=Array&UTM_LeadSource= Their initial effort created an ensemble model from existing models https://www.certara.com/wp-content/uploads/2019/09/Chelliah_2019_PAGE_immunotherapies.pdf. One of the take home points from our work is the importance of tumor cell functional plasticity and fibroblasts. We

note that neither are represented in this ensemble model generated by Certara. As mentioned in the discussion, current approaches rely on hand-curated mathematical models, where including each arc into the causal network is based on a discussion among domain experts. The model developed by Certara reiterates the point related to the problem with hand-curated models – they only include what experts think are important instead of letting the data speak for themselves. Furthermore, if hand-curated models are the current state-of-the-art in the immuno-oncology space, then we feel that this approach described in the manuscript is more data-driven and a significant advance. While we note that there still is some input into the process from domain experts that enters through specifying a blacklist, we feel that this seems almost negligible compared to a causal network specified entirely by hand. Ultimately, any mechanistic model that is generated must be validated experimentally, which we have done here. Specifically, we provide a time-course data showing that CCN4 exposure promotes an epithelial-mesenchymal-like transition in melanoma cells. In addition, CCN4 inhibits immune cell production of IFN-gamma, which is represented in the model as CCN4 promoting resting NK cells (or inhibiting active NK cells). CCN4 also has no effect on T cell proliferation, as predicted by the inferred causal network.

Reviewer #2 Comments:

The authors answered the concerns raised by this reviewer and two other reviewers in the response letter. The authors conducted additional experiments to validate the computational predictions. This reviewer could find the biological importance of CCN4 to the regulation of immune cell population changed associated with EMT in mouse model (by validation) and that digital flow cytometry has contributed for successful identification of immune cell populations from TCGA bulk cell datasets. I also understand that the authors made substantial efforts to improve the manuscript for this revision.

However, the whole manuscript is still not convincing to me.

R2: The biggest concern of this reviewer is the generality of this analysis method. A method of predicting cellular communication networks that combines digital flow cytometry and Bayesian network inference seems to have potential for wide use if this method is generally applicable to other types of bulk sequence datasets.

AR: The authors thank the reviewer for seeing the potential of this approach. Phenotypic screening approaches have the potential to identify proteins that alter the modelled phenotype. However if the particular protein is not well-characterized, as in the case of CCN4, a barrier for targeting a protein is identifying what it does beyond impacting the modelled phenotype in terms of altering the intercellular network. It was our intention to prototype an approach that can address this question in a data-driven manner.

R2: However, as I wrote in my last review, I'm still not sure how to choose CCN4. This impression is mainly due to the lack of information on (1) the CCN4 level (protein or mRNA) of the original TCGA dataset, and ... If CCN4 is not highly expressed in the datasets (BRCA and SKCM), or if there is no relationship between mRNA levels and malignancies (or prognosis) of the cancer, it is difficult to interpret this study focuses on CCN4-centric networks. Such data analysis is necessary at the beginning of the study.

AR: We apologize for not clarifying why we focused on CCN4. However in the interest of brevity and not repeating published work, we cite papers in the bottom paragraph in the left column on pg 2 that address this point. To elaborate on this point a bit more, we note that reinvigorating cell-mediated immunity to malignancy using immune checkpoint blockade has revolutionized the treatment of cancer. At least demonstrated using preclinical models, the response to immune checkpoint blockade depends on tonic IL12 signaling in the tumor microenvironment (TME) (see Garris et al. *Immunity* 49(6):1148-1161 (2018)). Consistent with but pre-dating that observation, our prior work (see Kulkarni et al. *Integr Biol* 2012 PMID 22777646) was based on the hypothesis that malignant cells evolve to secrete proteins that locally inhibit the response of CD4+ and CD8+ T cells to Interleukin-12 (IL12), given that oncogenesis is an evolutionary process involving repeated mutation and selection. In this study, we identified tumor-derived Cell Communication Network factor 4 (CCN4, previously known as WISP1) as a secreted inhibitor of immune response to IL12 using a phenotypic screening assay based on the B16F0 mouse melanoma model and a mass spectrometry-based secretome analysis. Of note, there are only about 500 papers related to CCN4/WISP1 that have been published since 1993, so the biology related to CCN4 is not well characterized. To clarify CCN4 biology, we published in *Klinke PLoS Comp Biology* 2014 (PMID: 24426833) that WISP1 (CCN4) is upregulated in essentially every tumor sample from patients with invasive breast cancer but not in normal tissue (p-value for significant difference between tumor and normal < 1e-15). In a follow-on mechanistic study, we show in *Klinke et al. Mol Biol of Cell* 2015 (PMID: 26224311) that disruption of adherens junction pathways, which occurs during invasion, increases WISP1 (CCN4) expression with an interlocked positive and negative feedback network motif. In relation to melanoma, we reported in *Deng et al. J Biol Chem* 2019 (PMID: 30723155), *Deng et al. Cell Mol Bioeng* 2020 (PMID: 32030107), and *Fernandez et al. EMBO Reports* 2022 (PMID: 35099839) that increased WISP1 (CCN4) correlates with worse outcome in melanoma and stimulates melanoma invasion and metastasis by promoting a process similar to the epithelial-mesenchymal transition. To identify the cell source of CCN4 within the tumor, we used scRNAseq data obtained from a cohort of 25 patients with melanoma. We found that CCN4 was expressed by malignant melanocytes in a similar proportion of patient samples compared to the TCGA SKCM primary melanoma cohort (9 of 25 versus 26 of 95 with Fisher Exact test p-value = 0.460 and odds ratio = 1.488) (see Fig. 1 in *Fernandez et al. EMBO Reports* 2022).

To improve the rigor of these initial studies, we expanded the focus beyond just CCN4 to test other secreted gene products identified using the workflow described in Kulkarni et al. *Integr Biol* 2012. These results are contained in a soon-to-be-released pre-print and shown in Figure R1 below. These additional secreted products considered include SPARC, a matricellular protein identified in the initial secretome analysis (Kulkarni et al. *Integr Biol* 2012) and thought to play a role in immunosuppression, and enriched components within extracellular vesicles secreted by B16F0 cells, namely DNMT3A and PTPN11 (Wu et al. *Pigment Cell Melanoma Res* 2017 PMID: 27930879). DNMT3A was selected as it was the most enriched transcript in B16F0 extracellular vesicles (EVs). Upregulation of DNMT3A enhances the suppressive properties of myeloid-derived suppressor cells and catalyzes the methylation of the IFNG promoter. PTPN11 was selected as both protein and mRNA were enriched in EVs and EV delivery of PTPN11 inhibited T cell proliferation in vitro.

Human transcriptomic and outcome data were used to provide a clinical context for these genes identified using the mouse B16F0 model for melanoma. Using data from the skin cutaneous melanoma (SKCM) arm of the Cancer Genome Atlas (TCGA), we analyzed data obtained at diagnosis from patients with primary melanoma and complete survival histories. A multivariate Cox proportional hazards model assessed the risk for progression based on expression of these different genes in patient tumor samples with age, sex, and stage as additional covariates (see Figure R1A). Among these seven covariates, only CCN4 expression predicted a significant change in hazard ratio (HR = 1.36, p-value = 0.032). To illustrate graphically the impact of CCN4 on overall outcome, a Kaplan-Meier survival analysis of this patient population stratified by CCN4 expression shows that increased CCN4 expression correlates with a worse outcome (p-value = 0.0013) (see Figure R1B). Hopefully that clarifies why we decided to focus on CCN4.

R2: the lack of information on ... (2) how to create "blacklists" and "whitelists". There is no such information in Method section. ... Creating blacklists and whitelists requires clear criteria that other researchers can follow and reproduce the method.

For these reasons, I am afraid to say that this manuscript lacks strong evidence of conclusions.

AR: We agree that creating "blacklists" and "whitelists" is an important point and is addressed extensively in almost three whole columns the "Results" section (top of right column on page 2 to middle of right column on page 3 that we have highlighted in red). Technical details, like the BIC formula, is given in the Methods section and the reader is also directed to the Result section ("The relationships among the nodes, or arcs, were represented by directed acyclic graphs inferred from the datasets using a process involving four steps, as detailed in the results section."). To further clarify our approach, we have now added a supplemental figure (shown below in Figure R2) that summarizes the approach for creating the blacklist and whitelists and cite it in the Results and Methods sections.

Figure R2. Analysis workflow for Bayesian Network Inference. The causal structure associated with cell-level networks were inferred from data using a four-step process. The four steps corresponded to specifying a "blacklist" based on prior information, generating an ensemble of potential arcs using 10 different structural learning algorithms, filtering potential arcs based on a trade-off between regression accuracy and model complexity as quantified by the Bayesian Information Criterion (BIC) to create a "whitelist", and learning the network structure and the corresponding parameter values using both the "blacklist" and "whitelist".

R2: In addition, data variation in each time point in Fig8 is too large to conclude the clear result.
AR: The results shown in Figure 8 are consistent with the dynamics reported previously, although with different cell lines. Here we use CCN4 KO variants of YUMM1.7 and B16F0 cells.

In prior work, we show that addition of recombinant mCCN4 (denoted as rmWISP1) to a CCN4 KO variant of B16F10 cells downregulates Cdh1 and increases Snai1 (see Figure R3 where we reproduce panel G of Figure 5 in Deng et al. JBiolChem 2019 and panel E of Figure 4 in Deng et al. Cell Mol Bioeng 2020). In Figure R3E shown above, the peak in Snai1 expression at 1 hour following the addition in medium containing CCN4 was also observed previously. Peaks in Snai2 and Zeb1 seem to come after the peak in Snai1. Snai1 is part of a genetic regulatory network that controls the epithelial-mesenchymal transition. It is not surprising that it's expression is not stable but dynamic. Furthermore, we show in Deng et al. JBiolChem 2019 that inducing expression of Snai1 in CCN4 KO cells phenocopies the metastatic phenotype.

Reviewers' Comments:

Reviewer #2:

Remarks to the Author:

The author has explicitly addressed concerns and uncertainties in this revision. I am satisfied with this content.

Response to Reviewer's Comments

Data-driven learning how oncogenic gene expression locally alters heterocellular networks.

David J. Klinke II, Audry Fernandez, Wentao Deng, Atefeh Razazan, Habibolla Latifizadeh, and Anika C. Pirkey

Submitted for review in Nature Communications (Manuscript ID: NCOMMS-21-07831D-Z)

Overall, we thank the reviewers for their service and their constructive criticisms on the manuscript. We feel that this revised paper presents, in a more comprehensive way, our findings related to inferring how genetic alterations associated with oncogenesis alters heterocellular networks within tissues.

Our responses to the specific comments are listed below in blue font. A red font is used to indicate the text that has been changed in this revised manuscript.

Reviewer #2 Comments:

The author has explicitly addressed concerns and uncertainties in this revision. I am satisfied with this content.

AR: The authors thank the reviewer for their service.